# The Role of Antimicrobial Resistance in Refractory and Recurrent Bacterial Vaginosis and Current Recommendations for Treatment

**DOI:** 10.3390/antibiotics11040500

**Published:** 2022-04-09

**Authors:** Christina A. Muzny, Jack D. Sobel

**Affiliations:** 1Division of Infectious Diseases, University of Alabama at Birmingham, Birmingham, AL 35233, USA; 2Division of Infectious Diseases, Wayne State University, Detroit, MI 48202, USA; jsobel@med.wayne.edu

**Keywords:** antimicrobial resistance, bacterial vaginosis, refractory, recurrent, treatment

## Abstract

Bacterial vaginosis (BV), the most common cause of vaginal discharge, is characterized by a shift in the vaginal microbiota from *Lactobacillus* species dominance to a diverse array of facultative and strict anaerobic bacteria which form a multi-species biofilm on vaginal epithelial cells. The rate of BV recurrence after therapy is high, often >60%. The BV biofilm itself likely contributes to recurrent and refractory disease after treatment by reducing antimicrobial penetration. However, antimicrobial resistance in BV-associated bacteria, including those both within the biofilm and the vaginal canal, may be the result of independent, unrelated bacterial properties. In the absence of new, more potent antimicrobial agents to eradicate drug-resistant pathogenic vaginal microbiota, treatment advances in refractory and recurrent BV have employed new strategies incorporating combination therapy. Such strategies include the use of combination antimicrobial regimens as well as alternative approaches such as probiotics and vaginal fluid transfer. Our current recommendations for the treatment of refractory and recurrent BV are provided.

## 1. Introduction

Bacterial vaginosis (BV) is the most common cause of vaginal discharge worldwide, with a global prevalence ranging between 23 and 29% across various regions around the world [1]. It is associated with multiple adverse health outcomes in women, including preterm delivery, pelvic inflammatory disease, and increased risk of acquisition of HIV and other sexually transmitted infections (STIs) [2,3,4]. BV is characterized by a shift in the vaginal microbiota from *Lactobacillus* species (spp.) dominance (i.e., *Lactobacillus crispatus*) to a diverse array of facultative (*Gardnerella vaginalis*) and strict anaerobic bacteria (i.e., *Prevotella* spp., *Atopobium vaginae*, *Sneathia* spp., etc.) which form a multi-species biofilm on vaginal epithelial cells [5]. The exact etiology of BV remains unknown although several hypothetical models have been published, centering around key BV-associated bacteria (BVAB) including *G. vaginalis*, *P. bivia*, *A. vaginae*, and *Megasphaera* spp. [6,7,8]. Epidemiological data strongly suggest that BV is an STI [9,10,11], although male partner treatment trials have yet to show an effect on reducing BV recurrence among women [12,13]. 

Despite the widespread availability of multiple oral and vaginal treatment options for BV belonging to the antibiotic classes of 5-nitroimidazoles (i.e., metronidazole, tinidazole, secnidazole) and macrolides (i.e., clindamycin) [14], the rate of recurrence after therapy can often be >60% [15]. This presents multiple emotional and economic challenges which can be a source of frustration for both women and clinicians alike who treat this common vaginal infection [1,16]. The BV biofilm itself likely contributes to refractory and recurrent disease after treatment by reducing antimicrobial penetration [17]. However, antimicrobial resistance (AMR) in BVAB, both within the biofilm and the vaginal canal, may also be the result of independent, unrelated bacterial properties. This article provides a narrative review of in vitro and in vivo data on antibiotic drug resistance in BVAB that may contribute to refractory and/or recurrent infection. It also provides our current recommendations for the treatment of these common infections in women, a topic not covered in detail in many national treatment guidelines [14,18]. 

## 2. In Vitro Data on Antibiotic Drug Resistance in BV-Associated Bacteria

Table 1 summarizes studies including in vitro data on antibiotic drug resistance in BV-associated bacteria. Nagaraja tested the in vitro antibiotic sensitivity of 50 strains of *G. vaginalis* to metronidazole and clindamycin [19]. In this study, 68% of the *G. vaginalis* strains were resistant to metronidazole while 76% were sensitive to clindamycin. Among the 17 *G. vaginalis* strains isolated from women with recurrent BV, 10 (58.8%) were resistant to metronidazole while all were sensitive to clindamycin [19]. Similarly, Li et al. evaluated the antimicrobial susceptibilities of metronidazole and clindamycin against 10 clinical isolates of *G. vaginalis* at both planktonic and biofilm levels [20]. Planktonic isolates showed significantly higher susceptibility (76.7% vs. 38.2%) and lower resistance (23.3% vs. 58.8%) to clindamycin than to metronidazole (*p* < 0.05 for both). In comparison to planktonic isolates, the minimum inhibitory concentration (MIC) of metronidazole was significantly higher for biofilm-forming isolates (7.3 ± 2.6 μg/mL vs. 72.4 ± 18.3 μg/mL; *p* = 0.005), the resistance rate was 27.3%, and the minimum biofilm eradication concentration (MBEC) was >128 μg/mL. The MIC of clindamycin was also higher for biofilm-forming isolates compared to planktonic isolates of *G. vaginalis* (0.099 ± 0.041 μg/mL vs. 23.7 ± 9.49 μg/mL; *p* = 0.034), the resistance rate was 27.3%, and the MBEC was 28.4 ± 6.50 μg/mL. The MIC and MBECs of clindamycin for biofilm-forming isolates of *G. vaginalis* in this study were lower than those of metronidazole. Overall, these data suggest that clindamycin may be better than metronidazole in vitro to eradicate *G. vaginalis* [19,20]. 

Petrina et al. subsequently evaluated the antimicrobial susceptibility of vaginal isolates of 605 BVAB and 108 lactobacilli to metronidazole, tinidazole, secnidazole, and clindamycin [21]. The MIC_90_ for secnidazole was similar to metronidazole and tinidazole for *Anaerococcus tetradius*, *A. vaginae*, *Bacteroides* spp., *Finegoldia magna*, *G. vaginalis*, *Mageeibacillus indolicus*, *Megasphaera*-like bacteria, *Mobiluncus curtisii*, *M. mulieris*, *Peptoniphilus lacrimalis*, *P. harei*, *Porphyromonas* spp., *P. bivia*, *P. amnii*, and *P. timonensis*. A proportion of *P. bivia* (40%), *P. amnii* (14%), and *P. timonensis* (58%) isolates were resistant to clindamycin with MIC values > 128 µg/mL. Metronidazole and secnidazole were superior to clindamycin for *Prevotella* spp., *Bacteroides* spp., *A. tetradius*, and *F. magna*. In contrast, clindamycin had greater activity against *A. vaginae*, *G. vaginalis*, and *Mobiluncus* spp. compared to the 5-nitroimidazoles [21]. Regarding vaginal lactobacilli, 100% of *L. crispatus* isolates, 96% of *L. jensenii* isolates, 19% of *L. gasseri* isolates, and 67% of *L. iners* isolates were susceptible to clindamycin (MIC ≤ 2) while the MIC_90_ for all lactobacilli tested was >128 µg/mL for the 5-nitroimidazoles. The authors concluded that secnidazole has similar in vitro activity against the range of BVAB compared to other 5-nitroimidazoles while sparing vaginal lactobacilli. 

The main resistance mechanism among clinically important BVAB and other anaerobic bacteria detected against macrolide antibiotics including clindamycin involves alteration of the antibiotic binding site by ribosomal methylation [22]. The ability of pathogenic bacteria to methylate the ribosomal target is coded for by erythromycin methylase genes (*erm* genes). Genes coding for 5-nitroimidazole resistance are referred to as *nim* genes [23]. These genes encode a nitroimidazole reductase enzyme which converts 4- or 5-nitroimidazole to 4- or 5-aminoimidazole, avoiding the formation of toxic nitro radicals that are essential for antimicrobial activity [24]. 

## 3. In Vivo Data on Antibiotic Drug Resistance in Women with Recurrent and Refractory BV

Table 2 summarizes in vivo data on antibiotic drug resistance in women with recurrent and refractory BV. The susceptibility of *G. vaginalis* isolates from 80 women with either single or multiple episodes of symptomatic BV before and after treatment with 2 g of oral metronidazole daily for 2–5 days has been examined [25]. The majority of pre-treatment isolates were susceptible to metronidazole, ranging between 88 and 100% based on the number of BV episodes. However, the number of susceptible isolates declined after the first (76–82%), second (53–82%), third (36%), and fourth (0%) rounds of treatment, respectively. There was also a trend towards higher MICs among resistant *G. vaginalis* isolates. Accordingly, the authors concluded that recurrent BV infections were more likely due to relapse than re-infection in this population of women [25]. An additional study of 117 women (27.4% of whom had BV) found that *G. vaginalis* biotypes 5 and 7 were most resistant to metronidazole [26]. Interestingly, while *G. vaginalis* is the most common BVAB found in most if not all cases of BV [27,28], *G. vaginalis* biotype 5 was predominantly associated with a healthy vaginal microbiota in this study, supportive of the hypothesis that *G. vaginalis* may be necessary but not sufficient for the development of BV [7]. In an earlier study of 95 women with BV (47 of whom received vaginal metronidazole for 5 days and 48 of whom received vaginal clindamycin ovules for 3 days), quantitative vaginal cultures were performed pre- and post-treatment for antimicrobial susceptibility testing. Of 1059 BVAB, <1% were resistant to metronidazole pre-treatment while 17% demonstrated clindamycin resistance. After treatment, no increase in metronidazole resistance was detected however 53% demonstrated resistance to clindamycin [29]. 

More recently, Bostwick et al. performed a case-control study of 326 age-matched women with and without BV using next-generation sequencing (NGS) to determine the prevalence of 14 pre-selected anti-microbial resistance (AMR) genes in each group [30]. They found more than a 4-fold-higher frequency of AMR genes in women with BV than in those without BV for macrolides (58.2 vs. 12.3%), lincosamides [a sub-class of the larger family of macrolide antibiotics] (58.9 vs. 12.3%), and tetracyclines (35.6 vs. 8.0%) (all *p* < 0.001). In this study *ermTR*, an AMR gene responsible for clindamycin resistance, was the most common gene present in both BV and non-BV specimens, although its prevalence in BV specimens was much higher (61.8%). In contrast, there was a low level of AMR gene identification (1.4%) for metronidazole (*nim* genes). One limitation of this study was that AMR gene findings were not linked to treatment outcomes. 

Deng et al. have also performed a meta-transcriptomic analysis of the vaginal microbiota of six women with persistent BV after treatment with metronidazole, comparing these results to those of 31 women with BV who were successfully treated [31]. They found that seven of eight clustered regularly interspaced short palindromic repeat (CRISPR)-associated (Cas) genes of *G. vaginalis* were highly upregulated in women with persistent BV suggesting that the CRISPR-Cas system may protect the vaginal microbiota against the DNA damaging effect of metronidazole. This finding has important implications for the development of novel therapeutic agents for women with persistent BV, as suppressing these genes may improve antibiotic therapy. 

Recently, whole-genome sequencing (WGS) has also been used to investigate the effect of metronidazole on the vaginal microbiota in five African American women with asymptomatic BV [32]. All subjects were tested for BV once every 2 months and received a 7-day course of oral multi-dose metronidazole for each BV episode over a 12-month time period. Despite treatment, none of the five women reverted to normal vaginal microbiota during the study; two were consistently positive for BV while three experienced intermittent infection. WGS analyses showed *Gardnerella* spp. to be the most highly abundant bacterial spp. associated with BV. Interestingly, after treatment with oral metronidazole, there was a decline in the relative abundance of *Lactobacillus* spp. and *Prevotella* spp. and an increase in the relative abundance of *Gardnerella* spp. over time (vaginal specimens were sequenced at four time points over the course of a 12-month time period in this study). The metagenome of all participants contained AMR genes; the most prevalent genes in this small cohort of women were *tetM* (associated with tetracycline resistance) and *IsaC* (associated with clindamycin resistance). Another resistance gene, *nimJ* (associated with metronidazole resistance), was detected in only a few of the specimens and at very low levels [32]. The authors of this study concluded that metronidazole may not be an effective treatment for women with asymptomatic BV and WGS may better inform the choice of antibiotics. 

In spite of all of the conflicting in vivo data described above, there is growing and convincing evidence of acquired AMR in BVAB, providing an answer to perplexed clinicians faced with high rates of clinical treatment failure. How to convert this conclusion into pragmatic therapeutic steps is as of yet unknown but implies the need to develop new antibiotics or better use existing agents, especially as combination regimens.

## 4. Treatment of Women with Refractory and Recurrent BV

Again, we emphasize that few professional medical societies responsible for publishing treatment guidelines address the clinical entity of relapsing or clinically unresponsive BV [18]. At minimum, a recent CDC recommendation suggested a maintenance regimen of twice-weekly metronidazole vaginal gel (0.75%) for 3 to 6 months aimed at mitigating BV relapse [15] but recognized that the benefits are only modest [14]. Similarly, little explanation is available in national guidelines as to the cause of BV treatment regimen failure or how to manage the patient.

### 4.1. Refractory BV Treatment

The approach to refractory and recurrent BV should be separate. Refractory BV is significantly less common than recurrent disease and in compliant patients is more likely to indicate AMR than a recurrent disease. It is not currently standard of care to obtain vaginal microbial samples for bacterial susceptibility testing in order to select a more effective antibiotic treatment regimen [14]; especially when no useful clinical guidance recommendations are available to guide in the selection of a regimen for the still symptomatic patient or even the partially symptomatic patient. Frequently, patients with refractory disease meeting both Amsel and Nugent criteria for BV, will acknowledge some reduced symptomatology, such as decreased odor or discharge or both after a course of treatment. It is tempting in asymptomatic women, but with persistent BV, not to recommend any further therapy, recognizing that rapid return of vulvovaginal symptoms is inevitable. A refractory response is more likely in the non-compliant patient and with the use of single-dose therapy rather than multi-dose therapy using 5-nitroimidazole medications. No guidelines exist guiding clinicians as to the next steps in the management of the refractory patient with persistent BV. Our approach, given the paucity of therapeutic options available, is to retreat the patient with two possible or consecutive steps.

First, the route of therapy should be switched (oral to vaginal or vice versa) but always with a multi-dose, non-abbreviated regimen (Figure 1). The second option is to switch the class of therapeutic drug (i.e., 5-nitroimidazole to clindamycin or vice versa). Although clinical studies exist documenting similar overall efficacy, several authors have reported a benefit in switching from metronidazole to 2% clindamycin cream or ovule administered over 7 days [33]. The explanation for this beneficial outcome is not clear. However, based upon the in vitro data described earlier in this manuscript, AMR of some BVAB to the antimicrobial drug classes is increasingly apparent. In addition, the microbial spectrum of these antimicrobial drugs, although largely similar, is not identical. Several BVAB strains are more sensitive to clindamycin including *Mobiluncus* spp., *G. vaginalis*, and *Atopobium vaginae* [21,33].

In contrast to randomized clinical trials, many if not all patients with refractory BV experience repeated exposure to metronidazole without ever receiving a single course of clindamycin. A successful outcome is not infrequent and welcomed by a jaded subpopulation of women. Failure to achieve a favorable clinical response precipitates several additional questions. Are all the 5-nitroimidazole drugs identical in efficacy? While some in vitro studies indicate minor advantages of tinidazole or secnidazole over metronidazole, no clinical data have emerged that women refractory to metronidazole are likely to respond to other 5-nitroimidazoles especially when the latter are prescribed for shorter regimens. Accordingly, we do not routinely recommend a 5-nitroimidazole drug switch for refractory or persistent disease. What options remain for refractory BV? Extending the duration of antimicrobial therapy from 7 to 14 days has not been shown to achieve higher cure rates [34].

Finally, does increasing the dose orally or concentrations vaginally of antibiotics offer any benefit in refractory BV cases? Once more, only a few studies have evaluated increasing oral or vaginal drug doses when faced with refractory episodes of symptomatic BV. The benefit of dose increase has been suggested by several authors [35,36]. However, since patient drug tolerance and toxicity preclude an increase in oral drug dose, this goal can be more easily accomplished by the vaginal route. The value of an increased vaginal dose of metronidazole was first suggested by Sanchez in 2004 [35]. More importantly, Sobel et al., when faced with women with oral metronidazole refractory disease with likely but unproven AMR, achieved some success in switching to high dose vaginal metronidazole 750 mg daily for 7 days. Unfortunately, a control arm was not available in this study [37]. Nevertheless, a beneficial role for substantially higher doses of vaginal metronidazole in women with likely, but unproven, AMR is suggested. Finally, if monotherapy with all available approved agents is ineffective, the use of combination therapy adding an anti-biofilm agent such as vaginal boric acid to an antibiotic simultaneously may be recommended, but once more there exists little supportive evidence except for data extrapolated from experience with recurrent BV [16].

### 4.2. Recurrent BV (RBV) Treatment

In principle, the management of RBV follows that of refractory BV, but BV recurrence after initial response to conventional therapy is likely due to factors other than AMR only. In particular, recurrence may be the result of reinfection from an asymptomatic male or female sexual partner and the likelihood and frequency of reinfection depend upon the patient population involved. While multiple sexual partners are recognized as a risk factor for initial BV infection, exposure to a single or the same sexual partner is a more important consideration in monogamous women with recurrent BV [38,39]. Sexual reinfection can only be excluded in celibate women. Whether oral sex plays a role is unknown but is not excluded as a contributory factor in women with RBV.

In managing women with relapsing BV and, possibly unrelated to reinfection, we have observed a subpopulation that is anything but homogenous. Some relapses occur within days or weeks after a course of antibiotics; in others, recurrence occurs after many months of no symptoms. Clinicians have long recognized that the absence of symptoms is not the primary consideration in women with RBV, in that some women during the “remission” period may still demonstrate all four Amsel criteria and similarly high BV Nugent scores. Frequently women with RBV who are asymptomatic immediately following conventional antimicrobials fail to achieve normal vaginal pH or fail to resolve the pre-therapy dysbiosis evident on wet mount microscopy. Yet other women with RBV relapse after weeks of presumed microscopy-determined eubiosis and a return of the vaginal pH to normal. Microbiota studies have not adequately addressed the role of vaginal dysbiosis in its varying forms with particular reference to prognostic microbiota criteria immediately following antibiotic therapy and longitudinally until recurrence occurs. Accordingly, treatment principles advocated in managing RBV currently rely exclusively on clinical studies often lacking comparative control groups and based upon limited available therapeutic options rather than implementing fact-based treatment principles.

A first reasonable step in managing RBV is once more to switch the class of antimicrobial from a 5-nitroimidazoles to 2% clindamycin for one week to achieve remission and hopefully long-term prevention of relapse (Figure 2) [14]. During this initial treatment phase, an effort should be made to eliminate host factors reported to be associated with RBV including the removal of intrauterine devices (IUDs), cessation of smoking, and avoiding unprotected sexual intercourse [14,40]. A popular next step is to initiate a maintenance prophylactic antibiotic regimen for 4 to 6 months. The most widely used regimen is twice-weekly vaginal metronidazole gel 0.75% which is moderately effective at achieving prevention of recurrence in approximately 70% of women with RBV [15]. The only adverse effect of the long-term use of vaginal metronidazole is frequent vulvovaginal candidiasis (VVC), occurring in 40–50% of women necessitating simultaneous administration of weekly prophylactic 150 mg oral fluconazole [16]. However, even in women responding to long-term vaginal metronidazole, high rates of BV recurrence follow rapidly with cessation of this antimicrobial regimen, implying the persistence of microbial pathogens in the vagina and, although unstudied, with a high likelihood of AMR [16]. This scenario is unfortunately not uncommon, and no treatment directives are available other than repeating the entire therapeutic process.

Nevertheless, a further step to resolve frequent BV recurrences not related to reinfection is forthcoming with a recent uncontrolled study by Surapaneni et al. in which women with RBV were treated with an initial combination therapy consisting of a 5-nitroimidazole orally 500 mg BID for 7 days together with simultaneous boric acid 600 mg daily per vagina [16]. The latter biofilm disrupter was prescribed for 30 consecutive days to achieve BV remission with high success. The protocol studied required further suppressive prophylaxis using maintenance twice weekly vaginal metronidazole gel for 5 additional months to complete an intensive 6-month regimen in women with frequently recurring and refractory RBV. This intensive and prolonged regimen achieved enhanced control and improved cure rates compared to historical controls but was still not without some BV recurrence in women following discontinuation of therapy [16]. With no new classes of antimicrobials in the pipeline, the immediate future for recurrent BV treatment looks dismal. Two products under study include a combination of vaginal boric acid and EDTA as maintenance prophylactic therapy for RBV [41] and Astodrimer 1% vaginal gel [42]. 

The role of alternative non-antimicrobial products in the management of recurrent and refractory BV remains equally un-reassuring. Probiotic use remains controversial in long-term reduction in BV recurrence [43,44] and it is not endorsed in the 2021 CDC STI Treatment Guidelines [14]. Although the use of a Lactin V (*L. crispatus* CTV-05) probiotic appears promising in the prevention of recurrent BV [45], this product is not yet commercially available. Remarkable results in a small study of five women with RBV were recently reported by Lev-Sagie et al. with vaginal microbiome transfer (VMT) from healthy female donors directly into the vagina of women with RBV immediately following conventional therapy with either 2% vaginal clindamycin for 7 days (n = 3) or 0.75% vaginal metronidazole gel for 5 days (n = 2) [46]. In this case series, four out of five of the women receiving a VMT transfer achieved full long-term remission at 5–21 months after VMT, defined as a marked improvement of symptoms, Amsel criteria, microscopic vaginal fluid appearance, and reconstitution of a *Lactobacillus*-dominated vaginal microbiota. The explanation for success was that initial partial eradication of a persistent resistant vaginal microbiota following antibiotic suppression that was subsequently infused with healthy exogenous vaginal microbiota allowed survival of the latter which became the dominant microbiota, eradicating residual resistant BVAB. More data from larger studies on this topic are needed.

It goes without saying that treatment of male sexual partners of women with recurrent BV has not been shown to be effective and is not recommended [12]. A recently completed multi-dose 7-day oral metronidazole treatment of women with RBV once more failed to reduce BV recurrence in women whose regular male sexual partner also received this treatment, although some benefit was forthcoming in compliant male partners, especially when accompanied by condom use [13].

## 5. Challenges in Conducting Research Studies of Women with Recurrent and Refractory BV and Future Directions

In spite of the global frequency of BV with numerous adverse health outcomes, clinical data related to both refractory and recurrent disease and causation thereof are surprisingly limited. Moreover, an understanding of the available in vitro and in vivo data related to AMR is remarkably deficient. Yet in spite of the paucity of data, there is no doubt that, contrary to initial studies, AMR exists among BVAB considered to be pathogens responsible for BV. In addition, evidence is accumulating that resistance exists in relation to both classes of drugs widely used for BV treatment (i.e., 5-nitroimidazoles and clindamycin).

However, this conclusion is only the beginning and not the end of the story. Virtually all past clinical efficacy studies over the last three decades failed to follow women longer than 30–40 days, ignoring BV recurrence; the FDA needs to require longer-term studies. In addition, details regarding past drug exposure are essential in BV treatment trials. When patients fail to respond or recur to therapy, we need to know why, and vaginal microbiota data should be available for susceptibility studies. Difficulties abound as long as we are unsure as to the critical pathogens to target. Needless to say, we need to include both planktonic and biofilm-based microorganisms in future studies, not to exclude the need to perform these studies in a polymicrobial environment.

Most importantly when performing traditional in vitro susceptibility studies in phase 2 and 3 drug efficacy studies, having selected a reasonable list of likely BVAB, we need to compare pathogens obtained pre- and post-drug treatment and correlate with clinical outcome (i.e., cure, refractory infection, or failure and recurrence). We lack data of this nature at present.

## 6. Conclusions

As clinicians are only at the beginning of the journey of investigation and when faced with patients with persistent vaginal dysbiosis (regardless of symptoms), therapeutic decisions are currently made without relevant patient-specific in vitro data. Thought should be given to the detection and measurement of pathogen-derived genetic markers of AMR moving forward. Clinicians, in the absence of new drugs for refractory and recurrent BV, should develop strategies for alternative treatment regimens, including the use of combination antimicrobial agents, probiotics, and/or vaginal fluid transfer, while recognizing the likelihood of AMR in managing women with these complicated infections.

## Figures and Tables

**Figure 1 antibiotics-11-00500-f001:**
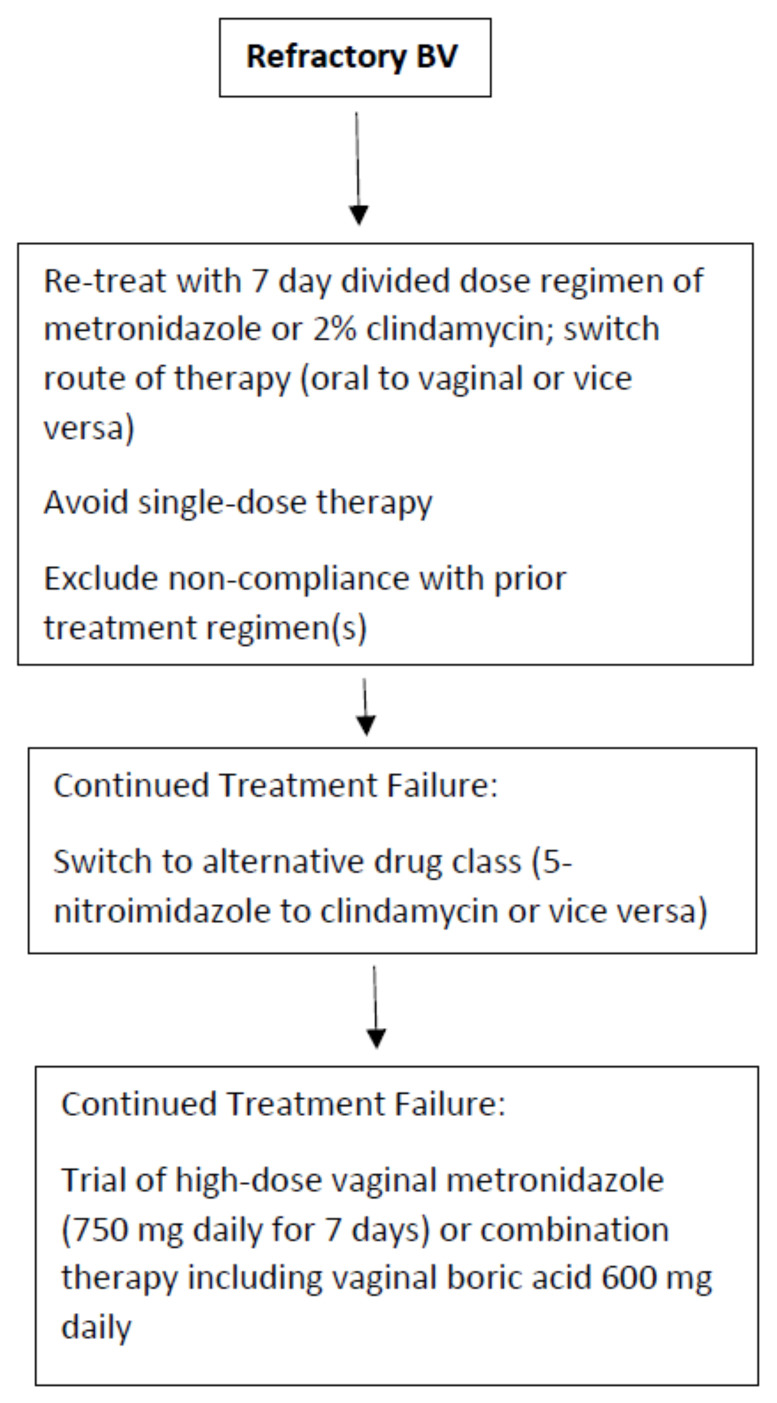
Recommended treatment algorithm for refractory BV.

**Figure 2 antibiotics-11-00500-f002:**
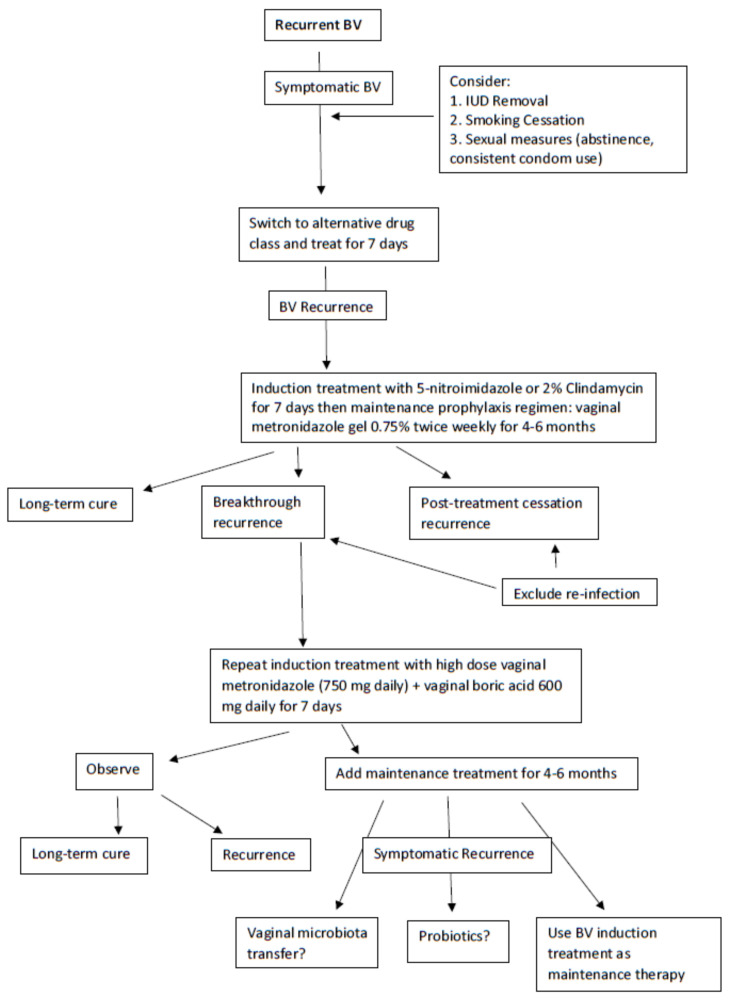
Recommended treatment algorithm for recurrent BV.

**Table 1 antibiotics-11-00500-t001:** In vitro data on antibiotic drug resistance in BV-associated bacteria.

First Author, Year	Bacterial Species, Number of Isolates Tested	Antibiotics Used in Susceptibility Testing	Results	Conclusions
Nagaraja, 2008 [19]	50 clinical isolates of *G. vaginalis*	MTZ, clindamycin	34 (68%) of isolates resistant to MTZ; 38 (76%) of isolates sensitive to clindamycin	Clindamycin is better in eradicating *G. vaginalis* than MTZ in vitro
Petrina, 2017 [21]	605 BVAB	MTZ, TDZ, SEC, clindamycin	MIC_90_ for SEC was similar to MTZ and TDZ for A. *tetradius*, *A. vaginae*, *Bacteroides* spp., F. *magna*, *G. vaginalis*, *M. indolicus*, *Megasphaera*-like bacteria, *M. curtisii*, *M. mulieris*, *P. lacrimalis*, *P. harei*, *Porphyromonas* spp., *P. bivia*, *P. amnii*, and *P. timonensis.* A proportion of *P. bivia* (40%), *P. amnii* (14%), and *P. timonensis* (58%) isolates were resistant to clindamycin with MIC values > 128 µg/mL. MTZ and SEC were superior to clindamycin for *Prevotella* spp., *Bacteroides* spp., *A. tetradius*, and *F. magna*. In contrast, clindamycin had greater activity against *A. vaginae*, *G. vaginalis*, and *Mobiluncus* spp. compared to 5-nitroimidazoles	More than a third of the *Prevotella* spp. were resistant to clindamycin SEC has similar in vitro activity against a range of BVAB compared to MTZ or TDZ. It also spares vaginal lactobacilli (data not shown)
Li, 2020	10 clinical isolates of *G. vaginalis*	MTZ, clindamycin at planktonic and biofilm levels	Planktonic isolates had greater susceptibility (76.7% vs. 38.2%) and lower resistance (23.3% vs. 58.5%) to clindamycin vs. MTZ (*p* < 0.05 for both) In comparison to planktonic isolates, the MIC of MTZ was higher for biofilm-forming isolates, the resistance rate was 27.3%, and the MBEC was >128 µg/mL. The MIC of clindamycin was also higher for biofilm-forming isolates compared to planktonic isolates, the resistance rate was 27.3%, and the MBEC was 28.4 ± 6.50 µg/mL	Clindamycin may be a better treatment option than MTZ for *G. vaginalis*, as it exhibits relatively higher susceptibility and lower resistance rates in vitro

Abbreviations: MTZ = metronidazole; MIC = minimum inhibitory concentration; MBEC = minimum biofilm eradication concentration; BVAB = BV-associated bacteria; TDZ = tinidazole; SEC = secnidazole.

**Table 2 antibiotics-11-00500-t002:** In vivo data on antibiotic drug resistance in BV-associated bacteria.

First Author, Year	Patient Population	Bacterial Species Tested, Antibiotics Used	Results	Conclusions
Bannatyne, 1998 [25]	80 women with single or multiple episodes of symptomatic BV pre- and post-treatment with 2 g oral MTZ daily for 2–5 days	*G. vaginalis* isolates; MTZ	88–100% pre-treatment isolates were susceptible to MTZ, based on the number of BV episodes The number of susceptible isolates after first (76–82%), second (53–82%), third (36%), and fourth (0%) rounds of treatment, respectively, declined	Recurrent BV infections were more likely due to relapse than re-infection
Aroutcheva, 2001 [26]	117 women, 27.4% of whom had BV	*G. vaginalis* isolates; MTZ	*G. vaginalis* biotypes 5 and 7 were most resistant to MTZ although biotype 5 was predominantly associated with a healthy vaginal microbiota (*p* = 0.0004)	No specific phenotype or genotype of *G. vaginalis* causes BV
Beigi, 2004 [29]	95 non-pregnant women with BV pre- and post-treatment (47 received vaginal MTZ for 5 days and 48 received vaginal clindamycin ovules for 3 days)	1059 BVAB; MTZ, clindamycin	Pre-treatment: <1% and 17% of BVAB were resistant to MTZ and clindamycin, respectively Post-treatment: no increase in MTZ resistance in BVAB although 53% were resistant to clindamycin	Treatment of BV with clindamycin is associated with marked evidence of antimicrobial resistance among BVAB
Bostwick, 2016 [30]	326 age-matched nongravid women of reproductive age with and without BV	Next-generation sequencing used to describe the complete vaginal microbiota and identify bacterial genes associated with resistance to a wide range of antibiotics	AMR genes were identified in all drug classes tested: macrolides 35.2%; lincosamides, 35.6%; tetracyclines, 21.8%; aminoglycosides (streptomycin, gentamicin and tobramycin), 5.2% each; 5-nitroimidazoles, 0.3%;triazoles, 18.7% There was more than a fourfold-higher frequency of AMR genes in pathogens from BV than from non-BV patients for macrolides (58.2 versus 12.3%), lincosamides (58.9 versus 12.3%) and tetracyclines (35.6 versus 8.0%), respectively	AMR genes were present in the majority of vaginal microbiomes of women with symptomatic BV
Deng, 2018 [31]	37 women with BV, of which 31 were successfully treated with MTZ	Meta-transcriptomic analysis of the vaginal microbiota was performed, comparing women who responded to BV treatment versus those who did not	7 of 8 clustered regularly interspaced short palindromic repeat (CRISPR)-associated (Cas) genes of *G. vaginalis* were highly upregulated in women with persistent BV	The CRISPR-Cas system may protect the vaginal microbiota against the DNA damaging effect of MTZ; suppressing these genes may improve the antibiotic therapy of BV
Ruiz-Perez, 2021 [32]	5 African American women ages 19–22 with asymptomatic BV at baseline followed over 1 year; women received oral MTZ for each BV episode during this timeframe	Whole-genome sequencing was used to determine changes in the vaginal microbiota among women with BV treated with MTZ	Despite treatment, none of the 5 women reverted to normal vaginal microbiota during the study; 2 were consistently positive for BV while 3 experienced intermittent infection *Gardnerella* spp. were the most highly abundant bacterial spp. associated with BV. After treatment with MTZ, there was a decline in the relative abundance of *Lactobacillus* and *Prevotella* spp. and an increase in the relative abundance of *Gardnerella* spp. over time The metagenome of all participants contained AMR genes	This study showed specific microbiota changes with treatment, presence of many AMR genes, and recurrence and persistence of BV despite use of MTZ

Abbreviations: MTZ = metronidazole; AMR = antimicrobial resistance.

## Data Availability

Not applicable.

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
