# Peer review of "The Role of Antimicrobial Resistance in Refractory and Recurrent Bacterial Vaginosis and Current Recommendations for Treatment"

_antibiotics, 2022, doi:10.3390/antibiotics11040500_

Round 1

Reviewer 1 Report

The authors aimed at their work to summarize  in vitro and in vivo data on drug resistance in BVAB that may contribute to refractory and/or recurrent infection. Topic is of much interest due to the the treatment of these infections in women. Overall work is interesting and very well written. I have some major comments, if these corrections are made, I believe the above work can be published to antibiotics.

Please add two tables to the review to describe the purpose and summarize the data, in the following sections:

  • One Table  in section summarize In vitro Data on Drug Resistance in BV-Associated Bacteria
  • One Table  in section summarize In vivo Data on Drug Resistance in Women with Recurrent and Refractory BV

- Line 10: change lactobacillus  to Lactobacillus species (Italic)

- Please check all names of the organisms and change to Italic

- Abstract and Conclusions: In order to entice readers, the Abstract and Conclusions should provide additional information.

References inside the text not as the style of Antibiotic Journal.

All manuscripts should be checked for typographical errors.

Author Response

Reviewer #1:

The authors aimed their work to summarize in vitro and in vivo data on drug resistance in BVAB that may contribute to refractory and/or recurrent infection. Topic is of much interest due to the treatment of these infections in women. Overall work is interesting and very well written. I have some major comments. If these corrections are made, I believe the above work can be published to Antibiotics.

  1. Please add two tables to the review to describe the purpose and summarize the data, in the following sections:

One Table  in section summarize In vitro Data on Drug Resistance in BV-Associated Bacteria

One Table  in section summarize In vivo Data on Drug Resistance in Women with Recurrent and Refractory BV

Author Response: The “In vitro Data on Drug Resistance in BV-Associated Bacteria” section has been updated to include a table summarizing the data on this topic (Table 1). The “In vivo Data on Drug Resistance in Women with Recurrent and Refractory BV” section has been updated to include a table summarizing the data on this topic (Table 2).

  1. Line 10: change lactobacillus to Lactobacillus species (Italic).

Author Response: This line has been changed to Lactobacillus species.

  1. Please check all names of the organisms and change to Italic.

Author Response: The names of all organisms throughout the manuscript have been checked and changed to italics, as needed.

  1. Abstract and Conclusions: In order to entice readers, the Abstract and Conclusions should provide additional information.

Author Response: We have added 2 new sentences to the end of the revised abstract which state: “In the absence of new, more potent antimicrobial agents to eradicate drug-resistant pathogenic vaginal microbiota, treatment advances in refractory and recurrent BV have employed new strategies incorporating combination therapy. Such strategies include the use of combination antimicrobial regimens as well as alternative approaches such as probiotics and vaginal fluid transfer.” In addition, the last sentence of the conclusion section of the manuscript has been revised to state “Clinicians, in the absence of new drugs for refractory and recurrent BV, should develop strategies for alternative treatment regimens, including use of combination antimicrobial agents, probiotics, and/or vaginal fluid transfer, while recognizing the likelihood of AMR in managing women with these complicated infections.”

  1. References inside the text not as the style of Antibiotics Journal.

Author Response: The references in the revised manuscript are in the style of the Antibiotics Journal, placed in square brackets [ ] before the punctuation.

  1. All manuscripts should be checked for typographical errors.

Author Response: This manuscript has been extensively checked for typographical errors.

Reviewer 2 Report

This review article summarizes antibiotic resistance in bacterial vaginosis (BV) from both in vitro and in vivo studies before discussing recommendation for (recurrent and refractory) BV treatment.

How about the potential of using prebiotics/synbiotics/postbiotics in management of BV?

Most guidelines state that treatment of BV is recommended for women with symptoms – non-pregnant and pregnant women. Perhaps the authors could also provide some insights on the management of BV among postmenopausal women.

Related literature:

10.1097/GME.0000000000001515

10.1139/w05-070

10.1128/JCM.40.6.2147-2152.2002

10.6118/jmm.2017.23.3.139

Other comments:

Please improve the following sentences for better clarity:

Page 1, Line 12: “While the BV biofilm itself likely…are discussed in this paper.”

Page 5, Line 202: “More importantly, Sobel et al, when faced with women with oral metronidazole refractory….in women with likely, but unproven, AMR is suggested.”

Please kindly ensure that scientific names are italicized.

Please avoid using too many question marks in the writing. Do consider improving the writing in a formal manner.

Author Response

  1. How about the potential of using prebiotics/synbiotics/postbiotics in management of BV?

Author Response: Although somewhat outside of the scope of this review, we currently make reference to the option of vaginal probiotics in lines 290-291, with reference to the Lactin V (L. crispatus CTV-05) trial that was recently published in the New England Journal of Medicine (reference 45). However, this product is not currently commercially available nor FDA-approved for women with recurrent BV and not listed in national treatment guidelines. Thus, we do not provide further depth on its use.

  1. Most guidelines state that treatment of BV is recommended for women with symptoms – non-pregnant and pregnant women. Perhaps the authors could also provide some insights on the management of BV among postmenopausal women.

Related literature:

10.1097/GME.0000000000001515

10.1139/w05-070

10.1128/JCM.40.6.2147-2152.2002

10.6118/jmm.2017.23.3.139

Author Response: Indeed this point is extremely controversial and exists in the face of a paucity of data, not with regard to treatment, but whether BV actually exists in postmenopausal women. Many authors question whether BV can occur in postmenopausal women not receiving estrogen replacement therapy (HRT). Pathogenic vaginal anaerobic bacteria are very estrogen-dependent, as much as vaginal lactobacilli. Certainly many postmenopausal women diagnosed with refractory or recurrent BV actually only have estrogen-deficient atrophic vaginitis and do not antibiotics but only to HRT. This topic is quite complicated and we have avoided discussing what many specialists consider to be a non-existent entity in this manuscript, which is already quite lengthy at 4,000 words. Perhaps this could be the focus of a future paper? 

Other comments:

Please improve the following sentences for better clarity:

  1. Page 1, Line 12: “While the BV biofilm itself likely…are discussed in this paper.”

Author Response: This sentence has been split into two sentences in the abstract and the revised introduction for clarity: “The BV biofilm itself likely contributes to recurrent and/or refractory disease after treatment by reducing antimicrobial penetration., However, antimicrobial resistance in BV-associated bacteria, including those, both within the biofilm and the vaginal canal, may be the result of independent, unrelated bacterial properties.”

  1. Page 5, Line 202: “More importantly, Sobel et al, when faced with women with oral metronidazole refractory….in women with likely, but unproven, AMR is suggested.”

Author Response: This sentence has been revised for clarity: “More importantly, Sobel et al, when faced with women with oral metronidazole refractory disease with likely but unproven AMR, achieved some success in switching to high dose vaginal metronidazole 750mg daily for 7 days. Unfortunately, a control arm was not available in this study.”

  1. Please kindly ensure that scientific names are italicized.

Author Response: All scientific names are italicized throughout the revised manuscript.

  1. Please avoid using too many question marks in the writing. Do consider improving the writing in a formal manner.

Author Response: We have intentionally added question marks only in select sections, to include only 3 sentences in paragraph 3 of the Refractory BV Treatment section of the manuscript. We request to leave these 3 sentences as is due to the controversary surrounding the discussion of the topic we are discussing.

Round 2

Reviewer 1 Report

 I believe the above work now can be published in antibiotics.

Reviewer 2 Report

Authors have addressed most of the comments and the readability of the manuscript has been improved substantially.